# Memory Induced by Recurrent Drought Stress in Chirca (*Acanthostyles buniifolius*)

**DOI:** 10.3390/plants14040555

**Published:** 2025-02-11

**Authors:** Tamara Heck, Gustavo Maia Souza, Marcus Vinícius Fipke, Rubens Antonio Polito, Andrisa Balbinot, Fabiane Pinto Lamego, Edinalvo Rabaioli Camargo, Luis Antonio de Avila

**Affiliations:** 1Department of Crop Protection, Federal University of Pelotas, Pelotas 96010-610, RS, Brazil; tamyheck@hotmail.com (T.H.); marfipke@gmail.com (M.V.F.); rubenspolito@gmail.com (R.A.P.); edinalvo_camargo@yahoo.com.br (E.R.C.); 2Department of Botany, Federal University of Pelotas, Pelotas 96010-610, RS, Brazil; 3Herbicide Research and Development, Syngenta Crop Protection, Lucas do Rio Verde 78455-000, MT, Brazil; andribalbinot@hotmail.com; 4Embrapa Clima Temperado, Pelotas 96010-971, RS, Brazil; fabiane.lamego@embrapa.br; 5Department of Plant and Soil Sciences, Mississippi State University, Starkville, MS 39762, USA

**Keywords:** climate change, chirca, priming effect, recurrent stress

## Abstract

To thrive as a successful weed in natural pastures, a plant must have not only highly competitive ability, but also the resilience to endure environmental stress and rapidly reclaim space once those stressors diminish and the other non-stress-tolerant plants die. *Acanthostyles buniifolius* [(Hook. ex Hook. & Arn.) R.M.King & H.Rob.], known as chirca, is a widely spread weed in South American natural pastures. It is known for its remarkable ability to withstand environmental stress and flourish in environments with prevalent stressors. The study evaluated the memory effect of water stress (drought) in chirca plants. The experiment was conducted in a greenhouse in a randomized block design with three replications. Treatments included Control = control plants without water deficit kept at 100% of the soil water-holding capacity (WHC); Primed plants = plants that were primed with water stress at 141 days after emergence (DAE) and received recurrent stress at 164 DAE; Naïve plants: plants that only experienced water stress at 164 DAE. To reach water stress, plants were not watered until the soil reached 15% of the soil’s WHC, which occurred ten days after water suppression in the priming stress and nine days after water suppression in the second stress. During periods without restriction, the pots were watered daily at 100% of the WHC. Primed plants exposed to water deficit better-maintained water status compared to the naïve plants; glycine betaine is an important defense mechanism against water deficit in chirca; naïve plants have a higher concentration of proline than plants under recurrent stress, demonstrating the greater need for protection against oxidative damage and needs greater osmotic regulation. Recurrent water deficits can prepare chirca plants for future drought events. These results show that chirca is a very adaptative weed and may become a greater threat to pastures in South America due to climate change, especially if drought becomes more frequent and severe.

## 1. Introduction

The plant environment is characterized by oscillating climatic factors, such as water availability or even successive periods of drought [1]. As a result, plants are driven by stressful stimuli to trigger a plethora of responses, such as stomatal closure and reduced transpiration, osmotic adjustments, and activation of antioxidant defenses to face the production of excessive reactive oxygen species (ROS) [2]. In addition to local responses in specific modules in plants, waves of ROS can play an important role, signaling a state of stress alertness from the stress perception site to the entire plant [1]. These waves travel through the plant during systemic signaling, changing the status of different plant cells and tissues along the way from a “normal” state to a state of “alert” or “stress”, triggering the inhibition of their development and directing energy resources to the metabolism of acclimation and defense [3,4].

During the developmental cycle, plants can be subjected to recurrent periods of exposure to drought, and often, only acclimated individuals can complete their growth cycle [5]. Exposure to consecutive environmental stresses can change the response of plants after each new stimulus, a phenomenon known as the “priming effect” or memory of stress [6]. Priming allows information to be stored to improve plant performance under new exposure to the stressor. The duration of this memory can vary from days to generations [7,8,9]. Therefore, stress memories can lead to a faster acclimatization response and increased tolerance in an upcoming water stress event [10,11] when compared to naïve plants. Naïve plants refer to plants not previously exposed to specific environmental conditions, treatments, pathogens, etc.

The plant can follow at least three different types of paths, from stimulus perception to the final stress response: “straightforward” response or STR (occurs immediately and without dependence on the history of stimuli previously received by the plant) or involving processes of memory of two types: “learning” (LRN) and “store/recall” (STO/RCL). However, it is not known whether the three response pathways are independent or connected [12]. In LRN memory, the repetition of the same stimulus modifies the intensity of the response, either negatively (“familiarization”) or positively (“sensitization”), induced by mild or severe stimuli, respectively [13]. In STO/RCL type memory, information emitted from a stimulus perception is stored inside the plant (storage function activated). This information remains in potential until, after an external or internal event that causes this information to be retrieved, it can influence metabolism in response to the subsequent stressful event; that is, the recall function is activated [14]. Thus, a first “priming” stimulus enables information storage to improve plant performance under a new stress condition [7,8,9].

In short, stress memory, also called stress imprinting, occurs when the plant that was previously subjected to a particular stressful event presents more efficient and faster responses during the second event of the same stress, which gives greater tolerance to recurrent events than naïve plants [15,16,17]. Stress memory can stabilize a plant community under frequent weather extremes to increase resilience and reduce plant mortality [5].

Different species have developed mechanisms to cope with water deficit stress, enabling their development [18,19]. In response, many plants synthesize and accumulate low molecular weight compounds, such as sugars, proline, and glycine betaine [20,21]. Proline accumulation is correlated with osmotic adjustment, carbon and nitrogen reserves used in post-stress recovery, ammonia detoxification, protein, membrane-stabilizing chaperone synthesis, and ROS scavenging [22]. Proline acts as an osmoprotectant that stabilizes membranes and maintains the conformation of cytosolic enzymes to survive water stress [23].

*Acanthostyles buniifolius* [(Hook. & Arn.) R.M.King & H.Rob.] belongs to the Asteraceae family and the Milleriaceae tribe and is one of the most troublesome weeds in natural pastures in the Pampa Biome regions, Southern Brazil [24]. It is a perennial plant with a branched shrubby size, reproduced only through seeds [25,26]. This rustic species is well adapted to poor and acidic soils, conditions frequently found in native field conditions in the Pampa Biome at Rio Grande do Sul state [26,27]. It is not used in animal feed, and without natural control, the increase in its dissemination causes the devaluation and reduction of the capacity to use the areas [26]. The Pampa biome in Brazil is characterized by significant trends in intense periods of drought [28,29]. In the state of Rio Grande do Sul, for instance, there is an increase in the frequency of droughts, especially in the southern half of the state, where rainfall is relatively low compared to the northern region [30]. Future forecasts show a decrease in rainfall in tropical South America. These changes are progressive, becoming stronger towards the end of the 21st century and at higher strength levels [31]. There are few plant species capable of dealing with extreme variations in water availability in the environment, and chirca is an example [24]. This plant’s adaptative characteristics and the fact that cattle avoid grazing it have led to its spread in this ecosystem. Climate change may exacerbate this issue, potentially making these weeds even more prevalent.

Within this context, the present work proposes to evaluate the possible memory effects in chirca plants submitted to recurrent stress due to water deficit, which may be at the base of its remarkable ability to occupy native areas. As a working hypothesis, we assumed that plants subjected to recurrent drought events in the second water stress situation would show higher stability in the maintenance of water status (water content and water potential closer to control plants) due to the priming effect on the mechanism of osmotic adjustment.

## 2. Material and Methods

### 2.1. Plant Material and Growth Conditions

To produce the seedlings for transplanting, they were sown in trays filled with sandy loam soil and placed in growth chambers at temperatures varying between 28 and 25 °C (day/night) and a 12 h photoperiod (800 μmol m^−2^ s^−1^). Ten days after emergence, two plants were transplanted into 8 L pots filled with sandy loam soil and transferred to greenhouse conditions.

The experiment was carried out in a completely randomized design, with three replications and three treatments: Control: plants without water deficit kept at 100% of the soil water-holding capacity (WHC); Primed plants: plants that were primed with water stress at 141 days after emergence (DAE) and received recurrent stress at 164 DAE; Naïve plants: plants that only experienced water stress at 164 DAE. To induce water stress, plants were not watered until the soil moisture level reached 15% of the soil’s water holding capacity (WHC). This condition was achieved ten days after water suppression during the priming stress period (first stress) and nine days after water suppression in the second stress period. During periods without water restriction, the pots were watered daily to maintain 100% of the soil’s WHC. The rehydration period last for five days. During the water restriction period, the pots were monitored by daily weighing (Figure 1).

Two sampling times were performed during the experiment to evaluate the physiological responses to the treatments in the second water restriction period (164 DAE) and after the rehydration period (Figure 1). Water potential (ѱw) relative to water contents (RWC) was measured immediately after sampling. Samples were stored in an ultra-freezer at −80 °C until analysis.

### 2.2. Biochemical Parameters

Hydrogen peroxide and lipid peroxidation: To estimate H_2_O_2_ and lipid peroxidation, leaf tissues (±0.25 g) were macerated with liquid nitrogen and homogenized with 0.1% (w:v) trichloroacetic acid (TCA) was used. The homogeneous mixture was centrifuged (12,000× *g*, 4 °C, 20 min), and the supernatant was used to determine the H_2_O_2_ content according to [32] Posso et al. (2020). Lipid peroxidation was determined [33] using thiobarbituric acid (TBA), which measures malonyldialdehyde (MDA) as the final product of lipid peroxidation. The amount of MDA TBA complex was calculated from the extinction coefficient (ε = 155 × 103 M^−1^ cm^−1^).

Photosynthetic pigments: To estimate chlorophyll A, chlorophyll B, and carotenoids, leaves (±0.02 g) were used. The samples were soaked in 7 mL of dimethylsulfoxide (DMSO) solution neutralized with 5% calcium carbonate, as [34] described, with some adaptations. Then, the test tubes were heated in a water bath at 65 °C for 4 h. After reaching room temperature, the absorbance of the homogenate was determined at wavelengths of 480, 649, and 665 nm [35].

Enzyme activity: The enzymatic activity was determined in leaves (± 0.25 g), as described by [36], which were ground in a mortar and pestle with liquid nitrogen containing 5% (w:v) of PVPP and potassium phosphate buffer 100 mM, pH 7.8, containing ethylenediaminetetraacetic acid (EDTA) and 20 mM sodium ascorbate. The homogeneous mixture was centrifuged at 12,000× *g* (for 20 min at 4 °C). An aliquot of the supernatant was used as a crude enzyme extract. An aliquot of the stratum was used to determine protein content, according to [37], using bovine serum albumin as a standard. The activity of superoxide dismutase (SOD; EC 1.15.1.1) was determined according to the inhibition of nitro-blue-tetrazolium (NBT) staining at 560 nm. The oxidation activity of ascorbate peroxidase (APX; EC 1.11.1.11) was determined by ascorbate oxidation at 290 nm.

Proline content and total soluble sugar: For the extraction of proline and total soluble sugars, plant tissues (±0.5 g) were ground in a mortar with 2 mL of MCW (methanol, chloroform, and water in a ratio of 12:5:3), as described by [38]. Proline dosage was determined by the ninhydrin method, according to [39], with some methodological adaptations. This method formed a biphasic phase, where 1 mL of the upper phase was collected and analyzed in a spectrophotometer at 520 nm. The absorbance obtained was compared with the standard curve for proline, and the results were expressed in µmol g^−1^ of fresh weight. Total soluble sugars were determined using the anthrone method [3].

Glycine betaine content: For the extraction of betaine glycine, plant tissues were ground in a mortar (±0.25 g), and 10 mL of deionized water was added, leaving the extract under stirring (230 rpm) for 24 h at 25 °C according to [40] Grieve and Grattan, (1983) with adaptations. The samples were filtered, an aliquot of 0.5 mL of the extract was added at a 1:1 ratio of sulfuric acid (H_2_SO_4_) 2 N, homogenized, and an aliquot of 0.5 mL was removed in polypropylene microtubes and kept on ice for 1 h. Added to this aliquot was 0.2 mL of potassium iodide (KI-I_2_), vortexed and stored at 0–7 °C for 16 h. After being thawed and homogenized in a vortex, the sample was centrifuged at 10,000 rpm for 15 min at 4 °C. The supernatant was carefully collected so that the precipitated betaine periodate crystals remained intact for washing with 3 mL of 1,2-dichloroethane. After 2 h with the crystals dissolved, a 2 mL aliquot was used to read a spectrophotometer at 365 nm.

### 2.3. Water Potential and Relative-Water Contents (RWC)

The water potential (Ѱw) analysis was performed on the fully expanded leaf of the middle part of the plant. The Ѱw was analyzed before dawn and noon using the Scholander Pressure Pump Chamber (SEC-3115-P40G4V, soil moisture, Santa Barbara, CA, USA). The variation of the water potential was determined by the formula: ΔѰw = Ѱw before dawn—Ѱw after dawn. The RWC was determined, as described by Barrs and Weatherley (1962), following the equation: RCW (%) = [(fresh mass − dry mass)/(water saturated mass − dry mass)] × 100. Sheet sections were removed from the middle part of the plant for each repetition. Then, these segments were weighed (fresh mass) and immediately placed in a plastic container filled with deionized water for 4 h to obtain the saturated mass of water. The saturated leaf material was dried at 60 °C to determine the dry mass until it reached constant weight.

### 2.4. Statistical Analysis

Statistical analyses were performed using R software 4.4.2 Before the analysis of variance, the data were tested for normality. When ANOVA identified significant differences, Tukey’s test was applied using 95% confidence intervals. Moreover, we used the Grammarly software (2024), which utilizes artificial intelligence, to proofread this manuscript.

## 3. Results

### 3.1. Water Relations

The leaf water potential results showed that primed plants showed a smaller reduction in both predawn and before dawn leaf water potential when compared to naïve plants (Figure 2A,C). This result is reflected in the relative water content values (Figure 2E), where primed plants showed higher RWC values than naïve plants. At the end of the rehydration period, the plants of all treatments recovered the values of RWC and leaf water potencies predawn and before dawn to values close to those of the control plants (Figure 2B,D,F).

### 3.2. Osmotic Adjustment Components

Primed plants had similar values of proline concentration and total soluble sugars (Figure 3A,E) when compared to naïve plants. The results of the glycine betaine concentration for naïve plants showed a 1.23 times higher concentration than that of primed plants (Figure 3C). At the end of the rehydration period, proline and glycine betaine had similar concentration values for all treatments (Figure 3B,D). However, for the variable total soluble sugars, the naïve plants showed a significant concentration increase compared to primed plants (Figure 3F). Primed plants and control plants presented similar values in the total soluble sugar concentration after the rehydration period’s end (Figure 3F).

### 3.3. Biochemical Parameters—Antioxidants

SOD activity showed that chirca primed plants had higher activity when compared to naïve plants (Figure 4A). After the end of rehydration, naïve plants and control plants presented similar values of SOD activity (Figure 4B). Naïve plants and primed plants stimuli showed increased hydrogen peroxide compared to control plants. However, compared to naïve plants and primed plants, they presented similar values of hydrogen peroxide (Figure 5C).

At the end of the rehydration period, naïve plants had a higher amount of hydrogen peroxide when compared to primed plants. However, primed plants and control plants presented similar values (Figure 5A) of hydrogen peroxide. These results reflect greater lipid peroxidation in naïve plants than in primed plants (Figure 5C). At the end of rehydration, lipid peroxidation values were similar for the treatments (Figure 5D). The activity of APX (Figure 5C,D), Chlorophyll A, Chlorophyll B (Figure 6A–D), and carotenoids (Figure 7A,B) in chirca plants, in both the period of water deficit and in the period of rehydration, was similar for all treatments.

## 4. Discussion

The lowest mean water potential values were recorded at midday (Figure 2B), reflecting a higher atmospheric demand for water vapor for chirca primed plants at midday. The water status of a plant directly influences the carbon assimilation capacity, regulated by stomatal conductance, so exposure to water deficit often causes a reduced photosynthetic rate due to greater stomatal resistance to reduce water loss through transpiration [40,41,42,43].

As hypothesized, primed chirca plants exhibited smaller changes in water potential and relative water content than naïve plants. The reduction in water potential and relative water content is a primary sign of water stress affecting water movement in the plant [44]. The reduction in the water potential is related to the increase in the transpiration rate, which occurs due to the high evaporative demand of the atmosphere [45]. The water potential at predawn is generally considered the best parameter to indicate the water condition of the plant. It may reflect the water potential in the rhizosphere region since both (plants and rhizosphere) remain in equilibrium during the night [46]. This effect is possibly due to a memory effect induced by priming, allowing these plants to respond faster and more effectively to stress. Similar effects were observed in clover plants subjected to two stress/recovery water deficit cycles, showing maintenance of water status and higher relative water content than plants subjected to only one stress cycle [42,43,47,48]. This demonstrates the occurrence of memory of stress. *Cistus albidus* plants subjected to recurrent stress showed interesting evidence for the physiological approach to memory, presenting osmotic adjustment mechanisms after exposure to a cycle of water stress and recovery [49].

The water potential of plants under water deficit is maintained through osmotic adjustment, that is, through the accumulation of organic solutes such as proline, soluble sugars, glycine betaine, and amino acids [50]. Osmotic adjustment can be considered one of the main plant adaptations resulting in water stress tolerance. Accumulating compatible osmolytes reduces the cell water potential below the external water potential, facilitating water movement into the cells [51,52].

In this study, primed plants had a lower concentration of glycine betaine when compared to naïve plants under later water holding (Figure 3C), possibly due to the priming effect of the first stress. The increase in glycine–betaine concentrations in plants under water deficit is probably associated with better absorption and transport of water from the soil to the shoot through osmotic adjustment, in addition to greater protection of the cell membrane, as well as protection against plant oxidative stress [53]. The protection of thylakoid membranes, which maintain photochemical efficiency in photosynthesis, is the primary function of glycine betaine [54]. Thus, this osmolyte maintains the water balance between the plant cell and the environment, stabilizing the macromolecules [55]. Similar results were found in papaya plants subjected to water deficit, showing an increase in the concentration of glycine betaine [56].

An increase in proline and soluble sugars concentrations was observed in chirca primed plants and naïve plants, indicating a greater capacity for the maintenance of leaf tissue water content, protein stability, and ROS levels. Increased proline and soluble sugar concentrations were also observed in *Hevea brasiliensis* plants subjected to water deficit and rehydration [45] and in rubber plants under water deficit [57]. After rehydration, chirca primed plants had similar total soluble sugar concentrations to control plants (Figure 3F), indicating that the priming effect possibly triggered a positive memory effect after rehydration. Likely positive responses of plants to future exposure to stress can also be expected, considering that they can become more resistant due to the acquisition of “memory”, a response that supports acclimatization or “hardening” [17].

The synthesis and degradation of proline are related to the processes of dehydration and rehydration, respectively. In the chloroplast, the enzyme P5CS (Pyrroline-5-carboxylate synthase) converts glutamate to proline when plants are dehydrated. At the same time, the enzyme PDH (proline dehydrogenase), which is responsible for the degradation of proline, becomes inactive. In our study, from the moment the plants were rehydrated, the proline concentration in the cells decreased (Figure 3). This may be related to the PDH enzyme that is activated again when rehydration occurs to convert proline to glutamate in the mitochondria and the P5CS enzyme is inactivated [58].

Furthermore, the increase in proline concentration may have resulted in a decrease in the activation of the antioxidant enzyme SOD in naïve plants (Figure 4B), showing higher activity in primed plants. Under stressful situations, proline concentration is usually increased in plants. This amino acid maintains the cell turgor pressure and protects enzymes and molecules from oxidation by reactive oxygen species (ROS) [6].

The action of the antioxidant enzymatic defense system through SOD activity (Figure 4A) in primed plants must have contributed to the reduction in the concentration of H_2_O_2_ and MDA (Figure 5A,C) when compared to naïve plants, indicating that chirca plants developed memory mechanisms that would support better performance under recurrent stressful events. Although each abiotic stimulus results in distinct responses, studies report that a first stressor event (*priming*) can prepare the plant for a subsequent event, resulting in a faster or more intense response [9,17,59,60,61].

After rehydration, naïve plants showed a higher level of MDA compared to the control (Figure 5). When the MDA content is high, it is indicative that the oxidative damage was intense, and the antioxidant system was not able to reverse the membrane damage by scavenging ROS. On the other hand, ROS waves in plants can play the role of an “alert” signal when there is a perception of a local disturbance, so this signal would change from a systemic location as the ROS waves propagate from cell to cell [62].

Photosynthetic metabolism is directly mediated by photosynthetic pigments, with chlorophyll A primarily responsible for enabling this interaction between the capture of solar energy. The process triggered within the chloroplasts, while chlorophyll B and carotenoids act as accessory pigments in the transfer of electrons to chlorophyll A [63,64] and exert a photoprotective function of the photochemical apparatus [65], predicting photo-oxidative damage to chlorophyll molecules [66]. Furthermore, the protection of thylakoid membranes, which maintains photochemical efficiency in photosynthesis, is performed by glycine betaine, its primary function [54]. The accumulation of glycine betaine found in chirca plants, when exposed to water deficit (Figure 3), may have contributed to the protection of photosynthesis efficiency.

Regarding photosynthetic pigments, there was no effect of treatments at both sampling times (after the drought stress and after the rehydration period) (Figure 6 and Figure 7), suggesting that the damage caused by oxidative stress with the formation of reactive oxygen, which is phytotoxic to plants, was not enough to oxidize these pigments or inhibit their synthesis [57]. Similar results were also found in *Hevea brasiliensis* plants when subjected to water deficit and rehydration, where no significant changes in the levels of chlorophyll A and B and carotenoids were identified [45].

## 5. Conclusions

Primed chirca plants show better water maintenance status when compared to naïve plants that receive only one form of stress. Naïve plants subjected to drought stress presented higher proline concentration than plants primed with drought stress, indicating a greater need for protection against damage caused by ROS and a greater need for osmotic regulation. The decrease in SOD activity in naïve plants exposed only to the later water deficit contributed to oxidative damage, as indicated by the increase in MDA levels.

Therefore, the exposure of chirca to recurrent stress resulted in a more efficient response to periods of drought, demonstrating a memory effect to recurrent water stress. These results show that chirca is a very adaptative weed and may become a greater threat to pastures in South America due to climate change, especially if drought becomes more frequent and severe.

## Figures and Tables

**Figure 1 plants-14-00555-f001:**
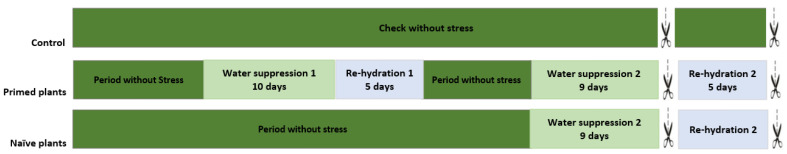
Distribution of treatments, where Control: plants without water deficit kept at 100% of the soil water-holding capacity (WHC); Primed plants: plants that were primed with water stress at 141 days after emergence (DAE) and received a recurrent stress at 164 DAE; Naïve plants: plants that only experienced water stress at 164 DAE. Water suppression 1 and 2 = first and second water suspension; Re-hydration 1 and 2 = rehydration period. Scissors indicate sampling time for further analyses (at the end of the water stress and at the end of the rehydration period).

**Figure 2 plants-14-00555-f002:**
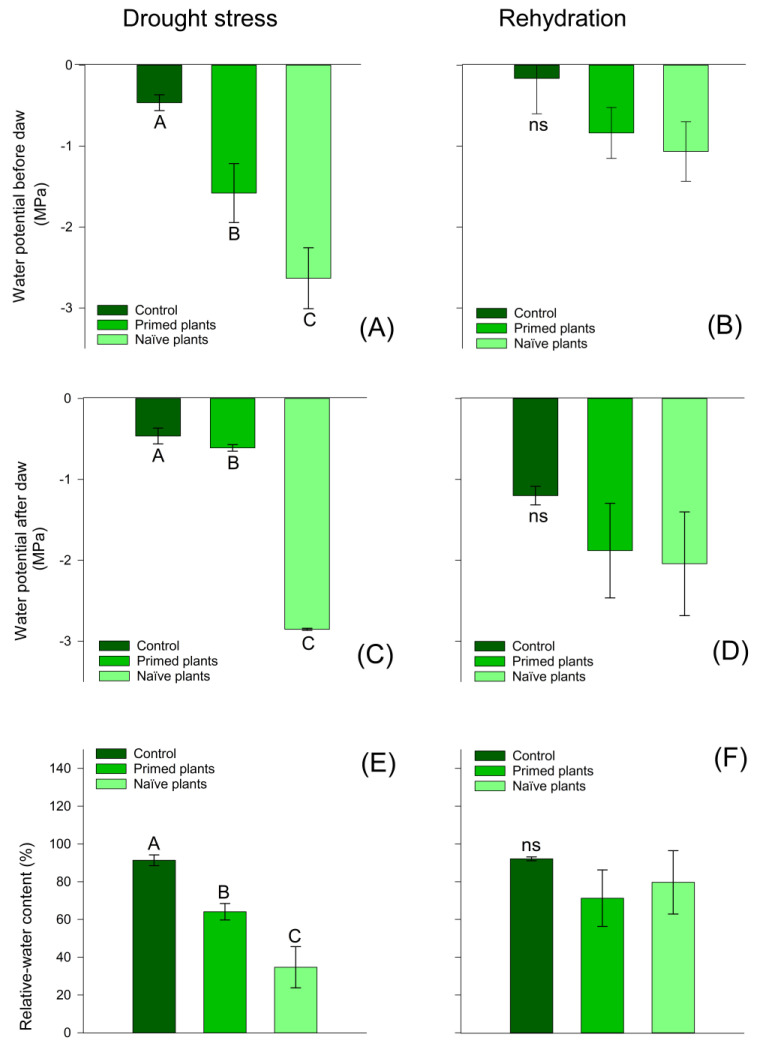
Effect of water stress on chirca leaf in water potential before dawn (**A**,**B**); in water potential after dawn (**C**,**D**); relative-water content (**E**,**F**); sampled after drought stress (**A**,**C**,**E**) and after the rehydration period (**B**,**D**,**F**). Error bars correspond to 95% confidence intervals (*n* = 3). Capital letters compare means between treatments; ns: not significant. Control: plants without water deficit kept at 100% of the soil water-holding capacity (WHC); Primed plants: plants that were primed with water stress at 141 days after emergence (DAE) and received recurrent stress at 164 DAE; Naïve plants: plants that only experienced water stress at 164 DAE.

**Figure 3 plants-14-00555-f003:**
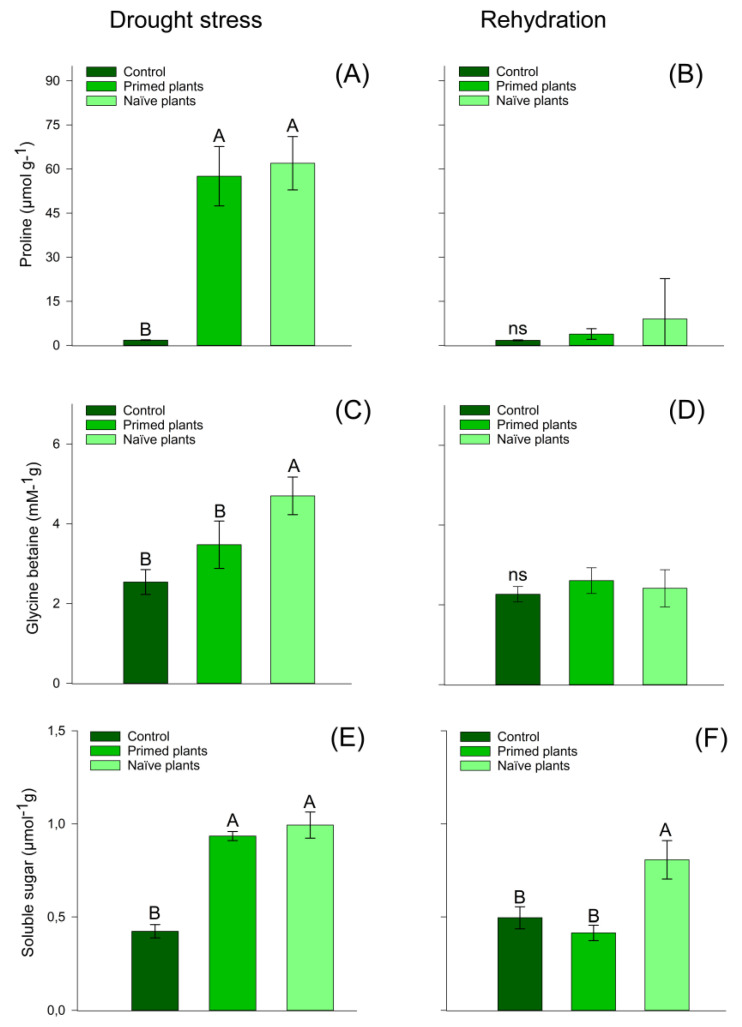
Effect of water stress in chirca leaf-proline content (**A**,**B**); glycine betaine content (**C**,**D**); soluble sugar (**E**,**F**); sampled after drought stress (**A**,**C**,**E**) and after the rehydration period (**B**,**D**,**F**). Error bars correspond to 95% confidence intervals (*n* = 3). Capital letters compare means between treatments; ns: not significant. Control: plants without water deficit kept at 100% of the soil water-holding capacity (WHC); Primed plants: plants that were primed with water stress at 141 days after emergence (DAE) and received recurrent stress at 164 DAE; Naïve plants: plants that only experienced water stress at 164 DAE.

**Figure 4 plants-14-00555-f004:**
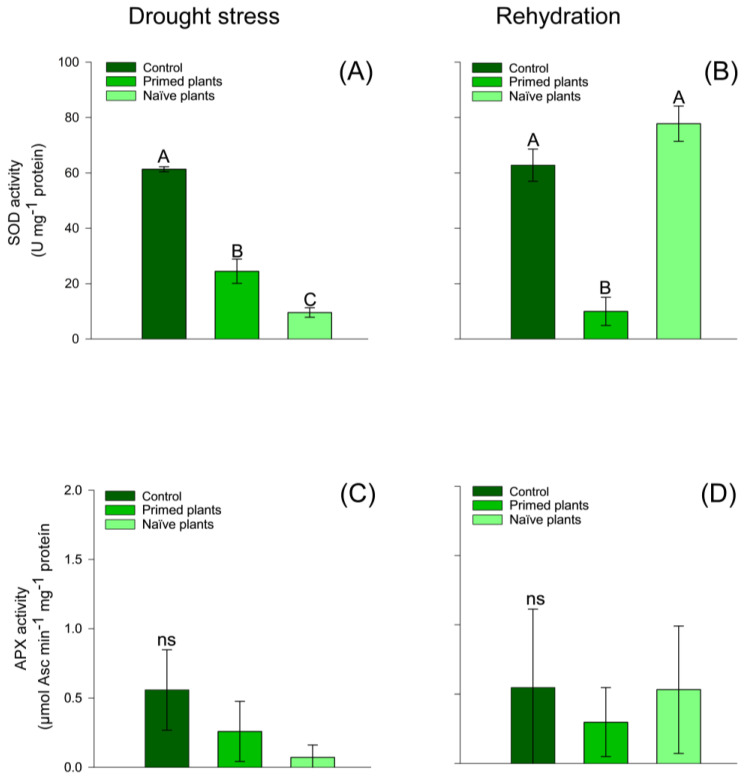
Effect of water stress on chirca leaf-SOD activity (**A**,**B**); activity of APX (**C**,**D**) in sampled after drought stress (**A**,**C**) and after the rehydration period (**B**,**D**). Error bars correspond to 95% confidence intervals (*n* = 3). Capital letters compare means between treatments; ns: not significant. Control: plants without water deficit kept at 100% of the soil water-holding capacity (WHC); Primed plants: plants that were primed with water stress at 141 days after emergence (DAE) and received recurrent stress at 164 DAE; Naïve plants: plants that only experienced water stress at 164 DAE.

**Figure 5 plants-14-00555-f005:**
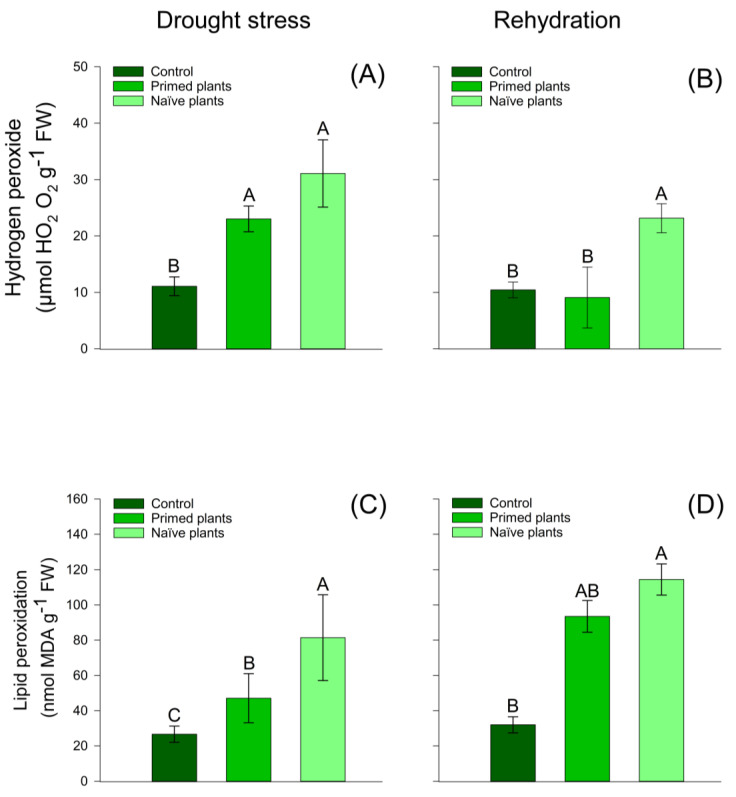
Effect of water stress in chirca on leaf-hydrogen peroxide (**A**,**B**); lipid peroxidation (**C**,**D**); sampled after drought stress (**A**,**C**) and after the rehydration period (**B**,**C**). Error bars correspond to 95% confidence intervals (*n* = 3). Capital letters compare means between treatments; ns: not significant. Control: plants without water deficit kept at 100% of the soil water-holding capacity (WHC); Primed plants: plants that were primed with water stress at 141 days after emergence (DAE) and received recurrent stress at 164 DAE; Naïve plants: plants that only experienced water stress at 164 DAE.

**Figure 6 plants-14-00555-f006:**
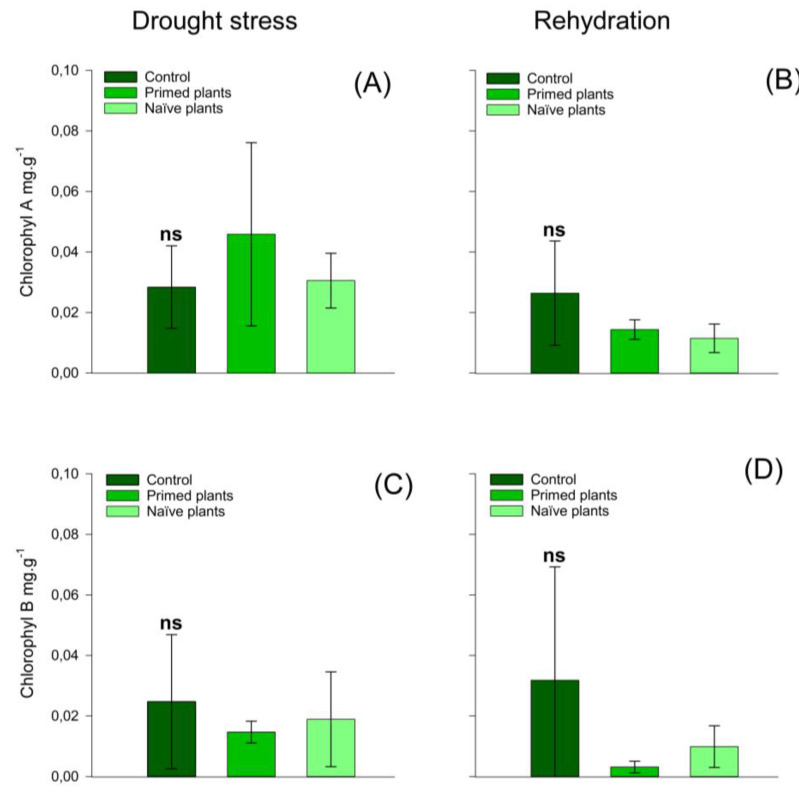
Effect of water stress in chirca on leaf-photosynthetic pigments, chlorophyll A (**A**,**B**); chlorophyll B (**C**,**D**); sampled after drought stress (**A**,**C**) and after the rehydration period (**B**,**C**). Error bars correspond to 95% confidence intervals (*n* = 3). Capital letters compare means between treatments; ns: not significant. Control: plants without water deficit kept at 100% of the soil water-holding capacity (WHC); Primed plants: plants that were primed with water stress at 141 days after emergence (DAE) and received recurrent stress at 164 DAE; Naïve plants: plants that only experienced water stress at 164 DAE.

**Figure 7 plants-14-00555-f007:**
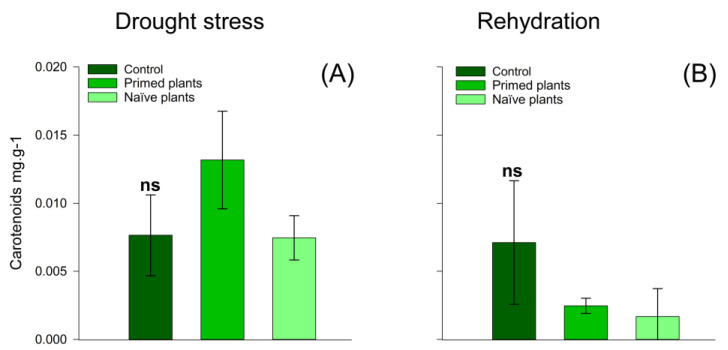
Effect of water stress on chirca leaf-carotenoid content sampled after the drought stress (**A**) and rehydration period (**B**). Error bars correspond to 95% confidence intervals (*n* = 3); ns: not significant. Control: plants without water deficit kept at 100% of the soil water-holding capacity (WHC); Primed plants: plants that were primed with water stress at 141 days after emergence (DAE) and received recurrent stress at 164 DAE; Naïve plants: plants that only experienced water stress at 164 DAE.

## Data Availability

The original contributions presented in the study are included in the article, further inquiries can be directed to the corresponding authors.

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
