# Peer review of "Memory Induced by Recurrent Drought Stress in Chirca (Acanthostyles buniifolius)"

_plants, 2025, doi:10.3390/plants14040555_

Round 1
Reviewer 1 Report
Comments and Suggestions for Authors
Acanthostyles buniifolius is a widely spread weed in South American natural pastures. It is acclaimed for its remarkable ability to withstand environmental stresses and flourish in environments with prevalent stressors. In this manuscript, the potted plants of A. buniifolius were treated with normal culture, early water stress, recurrent water stress, and late water stress, and biochemical parameters, water potential and relative-water contents of A. buniifolius plants, to identify a memory induced by recurrent water stress. However, there are many problems in the writing and data analysis of the manuscript:
1. The manuscript deals with A. buniifolius under water stress, but there are no phenotypic photos of plants before and after water stress. We know that the growth state and planting density of plants have a great impact on the stress resistance of plants.
2. Why are biochemical parameters and water potential and relative water contents determined at T1, T3 and T3, but not T2?
3. Figure 2 should be marked with T1, T3 and T4 correspond to the result description。
4. Figure 3C shows that glycine betaine in T3 is not significantly different from T1, but extremely significantly lower than T4. Figure 3F shows that soluble sugar in T4 is extremely significantly higher than T3. How to explain these results?
5. Figure 4A shows that SOD activity of T1 plants is significantly higher than that of T3 and T4, while T3 is significantly lower than that of T1 and T4 after rewatering. How to explain these results?
6. "The activity of APX (Fig. 5 C and D)" in line259, while the APX activity in the manuscript is Figure 4 C and D. line317 "... under later water holding (Fig. 4 C)" should be Figure 3C. There are many similar errors in this manuscript.
Author Response
Comments and Suggestions for Authors I
Acanthostyles buniifolius is a widely spread weed in South American natural pastures. It is acclaimed for its remarkable ability to withstand environmental stresses and flourish in environments with prevalent stressors. In this manuscript, the potted plants of A. buniifolius were treated with normal culture, early water stress, recurrent water stress, and late water stress, and biochemical parameters, water potential and relative-water contents of A. buniifolius plants, to identify a memory induced by recurrent water stress. However, there are many problems in the writing and data analysis of the manuscript:
- The manuscript deals with buniifoliusunder water stress, but there are no phenotypic photos of plants before and after water stress. We know that the growth state and planting density of plants have a great impact on the stress resistance of plants.
R: No photos were taken in this experiment. As mentioned in the methodology, two plants per pot were grown in 8L pots, a density that was defined due to the species' development characteristics.
- Why are biochemical parameters and water potential and relative water contents determined at T1, T3 and T3, but not T2?
R: Collections were only carried out at the end of the plant cycle, with T2 being just a pre-treatment.
- Figure 2 should be marked with T1, T3 and T4 correspond to the result description。
R: corrections made to the document
- Figure 3C shows that glycine betaine in T3 is not significantly different from T1, but extremely significantly lower than T4. Figure 3F shows that soluble sugar in T4 is extremely significantly higher than T3. How to explain these results?
R: Soluble sugars play an important role in maintaining the overall structure of the plant. The adaptive response, even after rehydration, can remain active temporarily, resulting in the maintenance of high sugar levels.
- Figure 4A shows that SOD activity of T1 plants is significantly higher than that of T3 and T4, while T3 is significantly lower than that of T1 and T4 after rewatering. How to explain these results?
R: The higher SOD activity in unstressed plants may reflect a functional and active antioxidant system maintaining the redox balance. However, plants under water deficit may show metabolic reduction and even enzymatic oxidative damage, resulting in lower SOD activity.
- "The activity of APX (Fig. 5 C and D)" in line259, while the APX activity in the manuscript is Figure 4 C and D. line317 "... under later water holding (Fig. 4 C)" should be Figure 3C. There are many similar errors in this manuscript
R: corrections made to the document
Others changes in the text
Títle: Before: Memory Induced by Recurrent Water Stress in Chirca (Acanthostyles buniifolius) Line 1,2;
Now: Memory Induced by Recurrent Drought Stress in Chirca (Acanthostyles buniifolius) Line 1,2;
Line 57,58,59: removed this phrase: The memorization process in plants can occur from plant hormones, pH changes, calcium-mediated signal transduction, gene expression, protein synthesis, and enzy-matic reactions (Demongeot et al., 2019).
Line 88: add one reference (Kishor et al., 2005)
Line 111, 112, 113, 114, 115, 116 and 117: The paragraph has been rewritten.
Line 118: rewritten the title : Material and Methods
Material and Methods line 125 to 135 has been rewritten
Figure 1 line 141: has been modified according to the suggested changes and the names of the treatments changed for better understanding by the reader
Line 142 to 147 has been rewritten
Line 196,199: the word “ daw” has been rewritten “dawn”
Line 212, 215,, 229, 231, 235, 248, 251, 255, 257, 298, 322, 335, 339, 355, 360, 392: the name of the treatment has been modified before “two water deficit stimuli (T3)” now “ primed plants”
Line 213, 215, 230, 234, 249, 251, 254, 257, 299, 323, 335, 354, 261, 367, 393: the name of the treatment has been modified before “ pre-treatment with water deficit (T4)” now “ naïve plants”
Figure 2 Line 220: has been modified and the description in line 221 to 227 has been rewritten
Fugure 3 Line 339: has been modified and the description in line 221 to 227 has been rewritten
Figure 4 Line 263: has been modified and the description in line 221 to 227 has been rewritten
Figure 5 Line 270: has been modified and the description in line 221 to 227 has been rewritten
Figure 6 Line 277: has been modified and the description in line 221 to 227 has been rewritten
Figure 7 Line 285: has been modified and the description in line 221 to 227 has been rewritten
Line 214,219, 230, 232, 233, 235, 249, 250, 253, 256, 257, 258, 259, 260, 323, 360, 267: the numbers and letters indicating the figures have been changed.
Line 383 to 386: has been rewritten
Line; 392 to 403: has been rewritten

Reviewer 2 Report
Comments and Suggestions for Authors
Consider discussing the specific physiological and biochemical pathways that enable chirca to withstand drought conditions
Explore the potential ecological consequences of chirca's resilience in natural pastures. How might this affect native plant species and overall biodiversity in those ecosystems?
Provide recommendations for land managers and farmers on how to deal with the increasing prevalence of chirca.
Suggest comparing chirca's resilience to other common weeds in similar environments.
Recommend long-term studies to assess how climate change, specifically increased frequency and severity of droughts, might impact chirca's growth and its interactions with other species over time.
What specific physiological changes occur in chirca during recurrent water deficit that enhance its drought resilience?
How does the presence of chirca affect the growth and survival of other plant species in the same ecosystem? Are there any documented cases of such interactions?
What role does genetic variation play in chirca's ability to thrive under water deficit conditions? Are there specific genotypes that are more resilient than others?
What are the implications of chirca's resilience for livestock health and productivity in pastures where it is prevalent? Are there any synergistic effects of water stress with other environmental factors (e.g., soil type, temperature) that might further influence chirca’s adaptability?
What future research directions would be most beneficial for understanding the long-term consequences of chirca's water stress memory on pasture ecosystems?
Comments on the Quality of English LanguageImprove english
Author Response
Comments and Suggestions for Authors II
Provide recommendations for land managers and farmers on how to deal with the increasing prevalence of chirca.
Suggest comparing chirca's resilience to other common weeds in similar environments.
R: The focus of the work is more related to aspects of physiological changes in Chirca plants, but in the future we could think about studies more related to pasture management in order to guide farmers
Recommend long-term studies to assess how climate change, specifically increased frequency and severity of droughts, might impact chirca's growth and its interactions with other species over time.
R: Future studies would be very important to understand how climate change could affect the development of the chirca and its interactions with the environment.
Consider discussing the specific physiological and biochemical pathways that enable chirca to withstand drought conditions
What specific physiological changes occur in chirca during recurrent water deficit that enhance its drought resilience?
R: In terms of physiological aspects, the parameters of photosynthesis and carotenoids were assessed, which showed no statistical difference between the treatments. In terms of structural changes in the plant, no alterations were observed with the naked eye that could increase the plant's resistance to drought, apart from the fact that the leaves wilted at the time of the water deficit and recovered quickly after the rehydration period.
How does the presence of chirca affect the growth and survival of other plant species in the same ecosystem? Are there any documented cases of such interactions?
R: Its presence can affect the development and growth of native plants in the Pampa Biome, as it is a hardy plant that is highly tolerant of poor soils and drought, as well as being a shrub that casts shade on surrounding plants. In addition, there is evidence of the release of allelochemical substances that inhibit the germination and development of neighboring species. However, there have been no specific studies on this elelopathic effect on chirca.
What role does genetic variation play in chirca's ability to thrive under water deficit conditions? Are there specific genotypes that are more resilient than others?
R: In this study, no genetic studies were carried out. There are no reports on genetic factors related to the ability to tolerate water deficit in chirca plants.
What are the implications of chirca's resilience for livestock health and productivity in pastures where it is prevalent? Are there any synergistic effects of water stress with other environmental factors (e.g., soil type, temperature) that might further influence chirca’s adaptability?
R: Because it is a highly invasive plant with no nutritional quality, the productivity of the cattle chain is negatively affected, with increased costs, a greater need for supplementation or more time to complete the production chain cycle. The ease of seed dispersal, adaptability to poor soils and periods of drought make it a plant more likely to thrive in these areas. The study only tested water deficit in isolation, and more studies are needed on the interaction of factors in order to conclude on cross-effects on the adaptability of chirca.
Suggest comparing chirca's resilience to other common weeds in similar environments.
What future research directions would be most beneficial for understanding the long-term consequences of chirca's water stress memory on pasture ecosystems?
R: studies related to possible transgenerational memory effects, seeking to understand whether in the long term this memory effect would be maintained or lost to future generations.
Other changes in the text
Títle: Before: Memory Induced by Recurrent Water Stress in Chirca (Acanthostyles buniifolius) Line 1,2;
Now: Memory Induced by Recurrent Drought Stress in Chirca (Acanthostyles buniifolius) Line 1,2;
Line 57,58,59: removed this phrase: The memorization process in plants can occur from plant hormones, pH changes, calcium-mediated signal transduction, gene expression, protein synthesis, and enzy-matic reactions (Demongeot et al., 2019).
Line 88: add one reference (Kishor et al., 2005)
Line 111, 112, 113, 114, 115, 116 and 117: The paragraph has been rewritten.
Line 118: rewritten the title : Material and Methods
Material and Methods line 125 to 135 has been rewritten
Figure 1 line 141: has been modified according to the suggested changes and the names of the treatments changed for better understanding by the reader
Line 142 to 147 has been rewritten
Line 196,199: the word “ daw” has been rewritten “dawn”
Line 212, 215,, 229, 231, 235, 248, 251, 255, 257, 298, 322, 335, 339, 355, 360, 392: the name of the treatment has been modified before “two water deficit stimuli (T3)” now “ primed plants”
Line 213, 215, 230, 234, 249, 251, 254, 257, 299, 323, 335, 354, 261, 367, 393: the name of the treatment has been modified before “ pre-treatment with water deficit (T4)” now “ naïve plants”
Figure 2 Line 220: has been modified and the description in line 221 to 227 has been rewritten
Fugure 3 Line 339: has been modified and the description in line 221 to 227 has been rewritten
Figure 4 Line 263: has been modified and the description in line 221 to 227 has been rewritten
Figure 5 Line 270: has been modified and the description in line 221 to 227 has been rewritten
Figure 6 Line 277: has been modified and the description in line 221 to 227 has been rewritten
Figure 7 Line 285: has been modified and the description in line 221 to 227 has been rewritten
Line 214,219, 230, 232, 233, 235, 249, 250, 253, 256, 257, 258, 259, 260, 323, 360, 267: the numbers and letters indicating the figures have been changed.
Line 383 to 386: has been rewritten
Line; 392 to 403: has been rewritten

Round 2
Reviewer 1 Report
Comments and Suggestions for Authors
The author answered all my questions and revised them as much as possible.
Author Response
There remain a few minor points that should be changed:
1. You talk about "4" treatments, but list only control, primed, and naive.
R: The number of treatments has been corrected in the line 133
The legends in Fig. 2 have "daw" instead of "dawn"
R: the correct figure has been add in the text
3. I would eliminate the paragraph that starts with "the water potential of plants is maintained through osmotic adjustment". That is not correct, and this paragraph is not really relevant to this study.
R: Thank you for your comment regarding the paragraph on osmotic adjustment. While we understand your suggestion to remove this section, we believe it is essential to retain this information. In our paper, osmotic adjustment was inferred from the measurements of proline, glycine betaine, and sugars. Although other elements, such as ions, also contribute to osmotic adjustment, the osmolytes we evaluated are widely recognized as classic indicators of this process.
Under water stress, various osmotically active molecules, including sugars, proline, glycine betaine, and organic acids, accumulate to regulate water relations. This accumulation causes a significant reduction in cellular osmotic potential, facilitating water uptake (endosmosis) and maintaining cell turgor. This mechanism, driven by the accumulation of compatible solutes/osmolytes, underpins the osmotic adjustment process. Notably, proline is regarded as one of the most critical osmolytes under water stress conditions (Sharma et al., 2019; Singh et al., 2015).
To support this, we have included references that highlight the role of these osmolytes as indicators of osmotic adjustment:
SHARMA, Anket et al. Phytohormones regulate accumulation of osmolytes under abiotic stress. Biomolecules, v. 9, n. 7, p. 285, 2019.
SINGH, Madhulika et al. Roles of osmoprotectants in improving salinity and drought tolerance in plants: a review. Reviews in environmental science and bio/technology, v. 14, p. 407-426, 2015.
Sakamoto, A., & Murata, N. (2002). "The role of glycine betaine in the protection of plants from stress: clues from transgenic plants." Plant, Cell & Environment, 25(2), 163–171.
It would have been a nice addition to this to have done some leaf gas exchange measurements.
R: The architecture of the leaf was one of the limitations to some of the analyses.

Reviewer 2 Report
Comments and Suggestions for Authors
Accepted
Author Response

(The authors gave the same response as above.)
